# Mice Lacking Connective Tissue Growth Factor in the Forebrain Exhibit Delayed Seizure Response, Reduced C-Fos Expression and Different Microglial Phenotype Following Acute PTZ Injection

**DOI:** 10.3390/ijms21144921

**Published:** 2020-07-12

**Authors:** Pei-Fen Siow, Chih-Yu Tsao, Ho-Ching Chang, Chwen-Yu Chen, I-Shing Yu, Kuang-Yung Lee, Li-Jen Lee

**Affiliations:** 1Graduate Institute of Anatomy and Cell Biology, College of Medicine, National Taiwan University, Taipei 10048, Taiwan; peifensiow@gmail.com (P.-F.S.); jacinta830512@gmail.com (C.-Y.T.); p310341@gmail.com (H.-C.C.); 2Department of Neurology, Chang Gung Memorial Hospital, Keelung 20401, Taiwan; imchenchwenyu@gmail.com; 3Laboratory Animal Center, College of Medicine, National Taiwan University, Taipei 10048, Taiwan; isyu@ntu.edu.tw; 4College of Medicine, Chang Gung University, Taoyuan 33302, Taiwan; 5Institute of Brain and Mind Sciences, College of Medicine, National Taiwan University, Taipei 10048, Taiwan; 6Neurobiology and Cognitive Science Center, National Taiwan University, Taipei 10617, Taiwan

**Keywords:** endopiriform nucleus, PTZ-induced seizure, hippocampus, neuronal activity, gene knockout mice

## Abstract

Connective tissue growth factor (CTGF) plays important roles in the development and regeneration of the connective tissue, yet its function in the nervous system is still not clear. CTGF is expressed in some distinct regions of the brain, including the dorsal endopiriform nucleus (DEPN) which has been recognized as an epileptogenic zone. We generated a forebrain-specific *Ctgf* knockout (Fb*Ctgf* KO) mouse line in which the expression of Ctgf in the DEPN is eliminated. In this study, we adopted a pentylenetetrazole (PTZ)-induced seizure model and found similar severity and latencies to death between Fb*Ctgf* KO and WT mice. Interestingly, there was a delay in the seizure reactions in the mutant mice. We further observed reduced c-fos expression subsequent to PTZ treatment in the KO mice, especially in the hippocampus. While the densities of astrocytes and microglia in the hippocampus were kept constant after acute PTZ treatment, microglial morphology was different between genotypes. Our present study demonstrated that in the Fb*Ctgf* KO mice, PTZ failed to increase neuronal activity and microglial response in the hippocampus. Our results suggested that inhibition of Ctgf function may have a therapeutic potential in preventing the pathophysiology of epilepsy.

## 1. Introduction

Seizures are the result of neuronal hyperactivity that causes involuntary behavioral changes, such as twitching of the limbs, while epilepsy is a chronic condition characterized by repetitive seizures and is one of the most common neurological disorders [1]. Various types of seizures occur in different age groups, leading to difficulties in diagnosis and treatments. Approximately 30% of patients do not respond well to the first-line anticonvulsants. Many of these patients require a combination of anticonvulsants but such treatments are still unable to effectively stop seizure recurrence [1,2]. The life expectancy is reduced up to two years in idiopathic (primary) epilepsy patients and 10 years in symptomatic (secondary) seizure patients [2]. Therefore, there is an urgent need for patients, clinicians and researchers to identify new drug targets and develop effective therapies. To achieve this goal, investigation of underlying mechanisms using mouse models in the field of experimental medicine is indispensable for successful drug development [3,4,5].

Connective tissue growth factor (CTGF), also known as Cyr61/CTGF/NOV (CCN) family protein 2 (CCN2), is a 36 to 38 kDa cysteine-rich secreted protein. CCN family members are multifunctional proteins that can facilitate a number of physiological functions such as growth factor binding and interactions among extracellular matrix and cell surface proteins [6]. CTGF is known to play important roles in the development and regeneration of various connective tissues [7,8]. However, the expression of CTGF is not restricted in the connective tissue but also in the nervous tissue, such as the olfactory bulb, cortical layer VIb and dorsal endopiriform nucleus (DEPN) in the brain [9]. The DEPN has been suggested as an epileptogenic site [10,11,12,13,14,15,16], implying that CTGF may play a role in the manifestation of epileptic seizures. 

To decipher the role of CTGF in the nervous system, we generated a forebrain-specific Ctgf knockout (*Emx1-Cre*; *Ctgf*^flox/flox^, Fb*Ctgf* KO) mouse line. In these mutant mice, the *Ctgf* gene is eliminated from the excitatory neurons in the forebrain structures [17]. The thickness of myelin sheath in the external capsule underneath the cortical layer VIb was reduced in Fb*Ctgf* KO mice, indicating a paracrine function of CTGF [17]. Nevertheless, the role of Ctgf in the DEPN, a proposed epileptogenic site, has not been examined. Glial cells constitute the majority of cell population in the nervous system and play important roles in its structural development and functional maintenance. The statuses of astrocytes and microglia are closely related to neuronal activity [18,19] and have been associated with epileptogenesis [20,21,22,23,24,25]. Since Ctgf is expressed exclusively in the excitatory neurons in the mouse brain [17], our Fb*Ctgf* KO mouse model provides an excellent opportunity to explore the interactions between neurons and glia in the pathophysiology of epilepsy [25].

In this study, we adopted an acute pentylenetetrazole (PTZ) paradigm to induce seizures in control and Fb*Ctgf* KO mice. PTZ is a bicyclic tetrazol derivative that suppresses GABAergic neurotransmission and has long been used for inducing convulsions and seizures in various animal models [26,27,28,29,30]. We scored PTZ-induced seizure behaviors and measured c-fos expression as well as glial responses in the mouse brain.

## 2. Results

### 2.1. Expression of Ctgf in the Mouse Brain

In the mouse brain, Ctgf is normally expressed in the olfactory bulb, cortical layer VIb, dorsal claustrum and DEPN but absent in the ventral claustrum or ventral endopiriform nucleus (Figure 1A). In the forebrain-specific *Ctgf* knockout (Fb*Ctgf* KO) mice, the expression of Ctgf protein was abolished (Figure 1B). Due to the unique expression pattern of Ctgf in the DEPN, we hypothesized that Ctgf may play a role in the manifestation of epileptic seizures. In this study, we examined the reactions of Fb*Ctgf* KO mice in a PTZ-induced seizure model.

### 2.2. PTZ-Induced Seizures in Mice 

Seizure scores were assessed through 30 min after PTZ administration. The latencies to generalized clonus (PTZ50, *p* = 0.41; PTZ60, *p* = 0.64) and tonic-clonic seizures (PTZ50, *p* = 0.40; PTZ60, *p* = 0.81) were similar between control and KO mice (Figure 2A,B). Some mice died after PTZ treatment within 30 min (PTZ50, *n* = 2 control and 7 KO mice; PTZ60, *n* = 3 control and 6 KO mice), their latencies to death were measured. There was no difference between groups (Figure 2C; PTZ50, *p* = 0.82; PTZ60, *p* = 0.09). Throughout the observation period, the overall seizure behaviors (excluding death) were not affected by the removal of *Ctgf* (Figure 2D). The seizure behaviors were then scored in a minute-by-minute manner. In control mice, after 50 mg/kg PTZ injection, the peak of seizure score occurred after 3 min, whereas this peak was delayed to 8 min post-injection in KO mice (Figure 2E). Two-way repeated measures ANOVA showed an interaction between genotypes and time after PTZ treatment between 3–8 min (F_(5,105)_ = 4.178; *p* = 0.008). This result suggested a delay in the seizure reaction in the Fb*Ctgf* KO mice and a reduction of sensitivity in PTZ-induced seizure paradigm in these mutants.

### 2.3. Expression of c-Fos in the Brain after Acute PTZ Treatment

We wondered if the discrepancy in the latency to the peak of the seizure score between genotypes was attributed to differential neuronal activity levels. Mice survived after acute PTZ injection were sacrificed at 2 h after PTZ treatment and we evaluated the index of neuronal activity by quantifying the number of nuclei positive to c-fos (a commonly used marker for neuronal activity) in the brain in these mice. Acute treatment with 30 mg/kg PTZ did not alter the expression of c-fos while 50 mg/kg PTZ injection induced c-fos expression in the cortex in both control and Fb*Ctgf* KO mice (Figure 3A). We measured the c-fos-positive nuclei in the upper cortical layers (layers II-III). In the motor cortex, in particular, robust c-fos expression was evident in both genotypes, indicating that the aberrant neural network activity might spread along the normal motor system. However, in the somatosensory and cingulate cortices, PTZ failed to induce significant increase of c-fos expression in Fb*Ctgf* KO mice (Figure 3B). In PTZ-treated mice, the expression of c-fos was elevated in layer VIb as other cortical layers.

Ctgf is normally expressed in the DEPN (Figure 1), we also checked the pattern of c-fos expression in the DEPN and its nearby regions, including the piriform, insular, perirhinal and entorhinal cortices (Figure 4 and Figure 5). Acute PTZ treatment (50 mg/kg) significantly raised the c-fos expression in the DEPN of control mice, but less prominent in mutant mice (Figure 4B). In the piriform cortex, PTZ elevated the number of c-fos-positive nuclei in both genotypes (Figure 4C).

In the insular, perirhinal and entorhinal cortices, acute PTZ treatment elevated c-fos expression in both control and KO mice, but the levels in KO mice were much lower than those in control mice (Figure 5). In our model, acute PTZ treatment did not increase the endogenous expression of Ctgf in the DEPN or cortical layer VIb in control mice.

The amygdaloid complex is involved in epileptogenesis [31] and it also receives direct projections from the DEPN [14]. Acute PTZ elevated c-fos expression in the amygdaloid complex, including the basolateral amygdala, central amygdala and basomedial amygdala (Figure 6A); however, the elevation was relatively moderate in Fb*Ctgf* KO mice (Figure 6B). In the hippocampus, a dramatic increase of c-fos expression was noted in the dentate gyrus (DG), hilus, CA1 and CA3 regions in PTZ-treated control mice. Surprisingly, PTZ-induced increase of c-fos expression in the hippocampus was absent in Fb*Ctgf* KO mice (Figure 6C,D).

We also quantified the expression of c-fos in the prelimbic and infralimbic cortices that also receive projections form the DEPN [32]. Similar to the pattern in the hippocampus, PTZ elevated the expression of c-fos in control, but not in Fb*Ctgf* KO mice (Figure 7).

### 2.4. Glial Reactions in the Hippocampus after Acute PTZ Treatment

The status of glial cells is closely associated with neuronal activity [18,19]. We next examined the features of astrocyte and microglial responses in the hippocampus following PTZ (50 mg/kg) treatment. Astrocytes were labeled by S100β-immunohistochemistry (Figure 8A). The densities of S100β-positive astrocytes were similar among the groups (Figure 8B). Glial fibrillary acidic protein (GFAP) is expressed in reactive astrocytes [33], we measured the percentage of GFAP-positive area as the index of astrocyte activation. However, this index was not altered following PTZ injection (Figure 9C,D).

We next checked the features of microglia in the hippocampus. Microglia were labeled by ionized calcium-binding adapter molecule 1 (Iba1)-immunohistochemistry (Figure 9A). The density of Iba1-positive microglia was not altered after acute PTZ treatment in both control and KO mice (Figure 9B). Since the morphology of microglia is highly associated with the function [19,34], we then examined morphometric features of individual microglia collected from the DG (Figure 9C–F). Notably, after PTZ injection, microglia in the control mice exhibited thicker segments (Figure 9C) and decreased lacunality (Figure 9D), an index of heterogeneity, signifying a sign of microglial activation [35]. We further checked the shape of microglia. The changes of microglial shape including the decrease of span ratio (Figure 9E) and increase of circularity (Figure 9F) were solely observed in the control mice. Together, these results showed that the features of microglial activation following PTZ injection were prominent in control but less evident in Fb*Ctgf* KO mice.

## 3. Discussion

In the present study, we demonstrated, the first time, the PTZ-induced seizures and cellular responses in mice lacking *Ctgf* in the forebrain structures, including the DEPN. We found delayed seizure response, reduced c-fos expression and lack of microglial activation following acute PTZ injection in Fb*Ctgf* KO mice. Our findings suggest a novel target for preventing the pathophysiology of epilepsy.

### 3.1. CTGF Involves in Various Fundamental Biological Process in the Living Organisms

CTGF is a small extracellular matrix (ECM)-associated fast turnover protein, critical for modulating multiple biological functions such as proliferation, differentiation, adhesion, migration, apoptosis and ECM remodeling [36,37,38]. CTGF has been well-known for its function of fibrogenesis and accounts for the fibrosis seen in Duchene muscular dystrophy (DMD) and Amyotrophic lateral sclerosis (ALS) mouse models. From a therapeutic point of view, neutralization of CTGF with monoclonal antibody has been proved effective and now under stage I or II clinical trial for idiopathic pulmonary fibrosis, pancreatic cancer and DMD [39].

CTGF could also integrate extracellular cues by modulating the bindings of growth factors with ECM component (e.g., integrin) with its conserved cysteine-rich functional domains and execute signaling transduction to the downstream molecules [40]. However, the knowledge accumulated from animal studies was limited and somewhat puzzling due to perinatal lethality of *Ctgf* KO mice [41] and transgenic mice overexpressing *Ctgf* in either kidney or heart that showed no fibrosis phenotypes [36]. In our model, the deletion of *Ctgf* could be executed by the expression of *Cre* which is driven by a designated promoter. This system could provide a powerful tool to decipher the role of *Ctgf* in a region- or cell type-specific manner.

### 3.2. Beyond Gliosis: The Emerging Role of CTGF in Regulating Brain Function

CTGF is normally expressed in the nervous system including the olfactory bulb, cortical layer VIb and DEPN. However, the function of CTGF in the brain is not well understood [9,17,42,43,44,45,46]. Based on the previous studies, CTGF may support critical function or even play beneficial role in the brain. For example, CTGF regulates the survival of newly generated neurons in the olfactory bulb [43]. Intriguingly, on the other hand, CTGF has been shown to account for the negative regulation of oligodendrocyte maturation in the developing brain [45]. In fact, CTGF may modulate the proliferation and differentiation of oligodendrocytes and control the growth and maturation of myelin sheath in a paracrine manner [17,42,45]. In addition, CTGF is involved in certain pathogenic conditions in the nervous system, such as the glial scar formation after traumatic brain or spinal cord injury [47]. In addition, CTGF has been reported upregulated in the spinal cord of ALS patients [48,49]; tuberous sclerosis complex [45], glioblastoma [50] and in the brain of Alzheimer’s disease [51]. In these diseases, however, the roles of CTGF are still unclear.

### 3.3. The Role of DEPN-Derived CTGF in the Regulation of PTZ-Induced Seizure Reactions

CTGF is expressed in the DEPN which is located deep in the piriform cortex and composed of densely packed multipolar cells [11]. DEPN has extensive mutual connections with nearby structures, including the piriform, insular, perirhinal and entorhinal cortices as well as the amygdaloid complex [14,15]. The anterior piriform/DEPN has the highest susceptibility to seizure initiation [10] and the extensive reciprocal projection pave the way for seizure propagation [13,14,15,52].

Following acute PTZ challenge, seizures were induced in both control and KO mice to a similar extent. However, the peak of seizures was delayed by 5 min in KO mice than that in control mice, indicating a higher threshold for seizure induction in these mutant mice. We are further examining this hypothesis using a PTZ-kindling paradigm.

In addition to the delayed onset of seizure-related behaviors in Fb*Ctgf* KO mice following PTZ treatment, the index of neural activity, c-fos expression, was less prominent in these mutants. In Fb*Ctgf* KO mice, PTZ failed to significantly elevate c-fos expression in the DEPN, implying a role of CTGF in modulating neuronal activity or excitability. However, since *Ctgf* gene has been removed before its native expression in these mutants, the role of Ctgf in the morphogenesis of DEPN neurons and network formation during neural development cannot be excluded.

DEPN neurons project to the entorhinal cortex, the gate of the hippocampal formation [52]. Interestingly, unlike the robust elevation of c-fos expression in PTZ-treated control animals, in Fb*Ctgf* KO mice, PTZ induced c-fos expressions were not significantly increased in all these regions, suggesting an DEPN-derived Ctgf-mediated gating mechanism in this DEPN-entorhinal-hippocampus pathway. On the other hand, DEPN neurons project to the perirhinal cortex which might serve as a gate-keeper to other cortical regions including the cingulate and insular cortices [14,15]. Along this DEPN-perirhinal-insular/cingulate pathway, PTZ significantly elevated c-fos expressions in KO mice yet not reached the control levels. These findings suggested another gating mechanism along the DEPN-perirhinal-insular/cingulate pathway that is also mediated by DEPN-derived Ctgf. However, the gating property of this system might be different from that in the DEPN-entorhinal-hippocampus pathway. Nevertheless, our findings suggested the propagation of PTZ-induced aberrant electrical activity is somewhat impeded in the DEPN, entorhinal and perirhinal cortices, the gates to the limbic structures including the hippocampus, amygdala and insular/cingulate cortex. Interestingly, Ctgf is not expressed in these brain regions but the DEPN, indicating the DEPN-derived Ctgf is mediating the gating mechanism. Further, PTZ-induced robust neuronal activity might be neurotoxic. Excitotoxicity induced by PTZ might impair the cognitive function by damaging the limbic structures [15]. In Fb*Ctgf* KO mice, PTZ-induced c-fos expression is relative moderate or insignificant, the consequential excitotoxicity might be spared in the limbic structures. Effort should be made to elucidate the structural and functional changes in the brains of PTZ-treated Fb*Ctgf* KO mice.

### 3.4. Glial Reaction and Epilepsy

Glial cells in the brain could respond to various stimuli and the status of these cells is highly associated with neuronal activity [18,19,20]. In this study, the index of neuronal activity, c-fos expression, was robustly elevated in the DG of PTZ-exposed control mice, but not in Fb*Ctgf* KO mice. Our results demonstrated that the acute reaction of DG microglia following PTZ treatment correlated well with the index of neuronal activity. Given the deletion of *Ctgf* was only operated in forebrain excitatory neurons, we believed the activation of microglia is resulted from PTZ-enhanced neuronal activity. Microglial activation may augment synaptic pruning, alter synaptic transmission and excitation/inhibition balance that lead to the development and aggravation of epilepsy [20]. Reactive microglia were noted in postmortem brain samples of patients with epileptic seizures and animal models of epilepsy [53]. In this regard, removal of Ctgf in the forebrain might be neuroprotective. We should observe the glial reaction following acute PTZ and chronic PTZ kindling paradigm at different time points.

### 3.5. Inhibition of CTGF May Have an Antiepileptic Potential

In this study, we characterized the PTZ-induced seizure reactions in mice lacking Ctgf in the forebrain. We proposed that following PTZ injection, DEPN-derived Ctgf may play a permissive role that allows the propagation of aberrant neural activity in the brain, especially the limbic structures. In contrast, deletion of Ctgf in the DEPN might alter the gating of neural signal propagation. Therefore, a possible strategy of limiting seizure-related neuropathy through the inhibition of CTGF function is worthwhile for further evaluation.

## 4. Materials and Methods

### 4.1. Animals

Mice of C57BL/6 background carrying *loxp*-floxed *Ctgf* sequence (*Ctgf*^flox/flox^) were crossed with *Emx1-Cre* mice in which the expression of Cre *reco*mbinase is driven by the promote*r* of Emx1 in forebrain excitatory neurons [54]. *Emx1-Cre*; *Ctgf*^flox/flox^ mice were generated and designated as forebrain-specific *Ctgf* knockout (Fb*Ctgf* KO) mice [17]. In this study, Fb*Ctgf* KO mice and their littermates (*Emx1-Cre*; *Ctgf*^+/+^, controls) were used. Animals were bred and kept in the Laboratory Animal Center of the College of Medicine, National Taiwan University, under a 12-h light/dark cycle (lights on at 08:00) with free access to food and water. Genotypes of the mice were examined using PCR-based protocol [17] and animals were weaned at postnatal day 28. All animal procedures were approved by the Institutional Animal Care and Use Committee of the College of Medicine, National Taiwan University (approval code: 20170291, approved on 23 September 2019).

### 4.2. Pentylenetetrazole (PTZ)-Induced Seizure Model

Adult male control and Fb*Ctgf* KO mice aged 3–5 months were intraperitoneally injected with different doses of PTZ (30, 50 or 60 mg/kg) or vehicle (normal saline) during 13:00–15:00. The behaviors of mice were recorded by a camera in front of the box for half an hour. The seizure profile was scored by experienced examiners blind to the genotypes. Normal locomotor activity was scored as 0, hypoactivity was 1, partial clonus was 2, generalized clonus was 3 and tonic-clonic seizures was 4 [55]. Two hours after PTZ injection, mice were sacrificed for histopathological examinations.

### 4.3. Immunohistochemistry

Two hours after saline or PTZ injection, mice were overdosed with 150 mg/kg sodium pentobarbital and transcardially perfused with 0.1 M PBS, followed by 4% paraformaldehyde. Whole brains were then harvested and postfixed overnight in the same fixative. Brain sections of 30 μm thick were cut using a vibratome (VT1000, Leica Biosystems, Wetzlar, Germany), reacted with 1% H_2_O_2_ to block the endogenous peroxidase activity and transferred to a PBS-based blocking solution containing 4% normal goat serum (Abcam, Cambridge, UK), 4% bovine serum albumin (Abcam, Cambridge, UK) and 0.4% Triton X-100 (Merck, Darmstadt, Germany). After blocking, sections were incubated with primary antibodies, such as goat anti-CTGF (1:1000; Santa Cruz Biotechnology, Santa Cruz, CA, USA), rabbit anti-c-fos (1:1000, Cell Signaling, Danvers, MA, USA), rabbit anti-S100β (1:1000, Abcam), rabbit anti-GFAP (1:1000, Abcam) or rabbit anti-Iba1 (1:1000, GeneTex, Irvine, CA, USA) antibodies in 10% blocking solution overnight. After washing, the sections were incubated with biotinylated secondary antibodies (1:500, Jackson ImmunoResearch Laboratories, West Grove, PA, USA) and avidin-biotin peroxidase complex (ABC kit, Vector Labs, Burlingame, CA, USA). Finally, sections were reacted with 3,3′-diaminobenzidine (with 0.01% H_2_O_2_ in PBS) and mounted. For control experiments, we omitted the use of primary antibodies and the immunoreactive signals were neglectable.

### 4.4. Cell Density Quantification

The densities of c-Fos-positive nuclei, S100β-positive astrocytes and Iba1-positive microglia were quantified by measuring the number of cells within a counting frame (150 μm × 150 μm) in given brain regions using the ImageJ software (NIH, Bethesda, MD, USA). To evaluate the index of astrocyte activation, the relative area of GFAP-positive signal in the counting frames was quantified.

### 4.5. Reconstruction and Morphometric Analysis of Microglia

Iba1-immunostained microglia were examined and image stacks were taken under a microscope equipped with the Stereoinvestigator system (Microbrightfield Bioscience, Williston, VT, USA) at 40× magnification using 1 μm z-steps. Microglia with definite cell bodies and processes were reconstructed and analyzed with FracLac plugin for ImageJ. Skeleton and fractal analyses were performed as described [56].

### 4.6. Statistical Analyses

All data were subjected to two-way ANOVA using SPSS (IBM Corporation, Armonk, New York, USA) to examine the interaction between treatment and genotype. When the *p-*values of ANOVA reached statistically significant (*p* < 0.05), *t*-tests with Bonferroni correction were performed to evaluate which experimental groups presented significant effects. Values are represented as the mean ± standard error of mean (SEM).

## Figures and Tables

**Figure 1 ijms-21-04921-f001:**
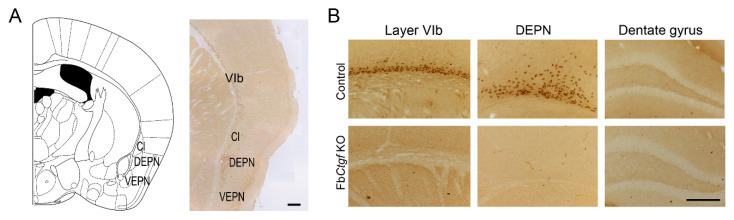
Connective tissue growth factor (Ctgf) protein is expressed in distinct regions of the brain such as the cortical layer VIb, dorsal claustrum and dorsal endopiriform nucleus (DEPN), but not in the ventral claustrum (Cl) or ventral endopiriform nucleus (VEPN) (**A**). The native expression of Ctgf protein was abolished in forebrain-specific *Ctgf* knockout (Fb*Ctgf* KO) mice (**B**). Both bars = 200 µm.

**Figure 2 ijms-21-04921-f002:**
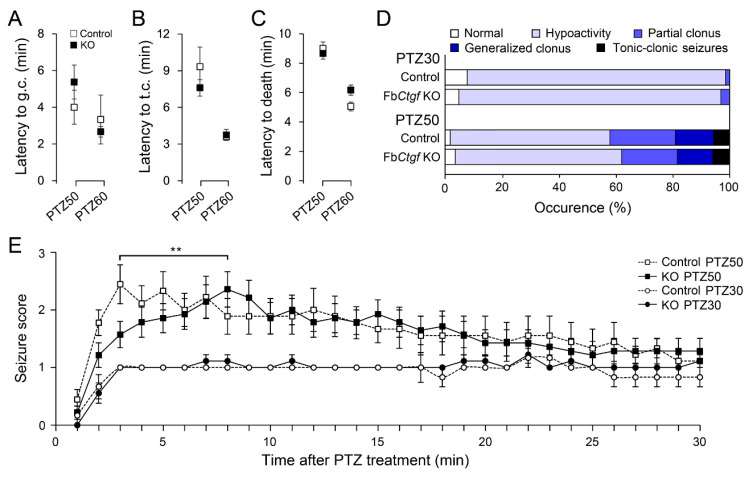
Behavioral responses after acute pentylenetetrazole (PTZ) treatment. Control and Fb*Ctgf* KO (KO) mice were subjected to PTZ injection (30 mg/kg, PTZ30; 50 mg/kg, PTZ50; 60 mg/kg, PTZ60). The latencies to generalized clonus (g.c.) (**A**), tonic clonic (t.c.) seizures (**B**) and death (**C**) were measured. During the 30 min observation period, the highest seizure reaction of each mouse was noted (**D**). The seizure behaviors were scored in a minute-by-minute manner (**E**). For PTZ30 group, *n* = 6 control and 9 KO mice; for PTZ50 group, *n* = 10 control and 20 KO mice; for PTZ60 group, *n* = 6 control and 8 KO mice. Normal locomotor activity was scored as 0, hypoactivity was 1, partial clonus was 2, generalized clonus was 3 and tonic-clonic seizures was 4. Results are mean ± SEM. Asterisks indicate significant difference (** *p* < 0.01).

**Figure 3 ijms-21-04921-f003:**
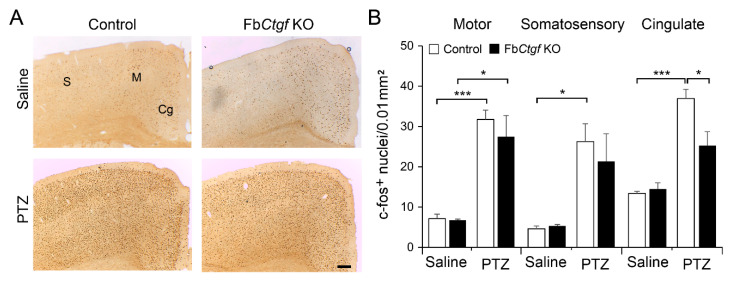
Expression of c-fos in the neocortex. Two hours after 50 mg/kg PTZ injection, animals were sacrificed for c-fos immunohistochemistry. The expression of c-fos was evident in the somatosensory (S), motor (M) and cingulate (Cg) cortices (**A**). Within these regions, the density of c-fos-positive nuclei was quantified (**B**). For saline group, *n* = 3 per genotype; for PTZ group, *n* = 5 per genotype. Results are mean ± SEM. Asterisks indicate significant differences (* *p* < 0.05; *** *p* < 0.001). Bar = 200 μm.

**Figure 4 ijms-21-04921-f004:**
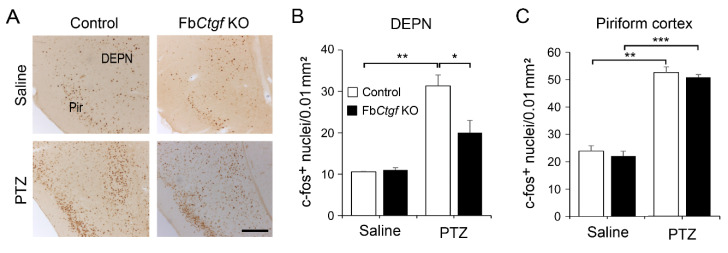
The expression of c-fos in the dorsal endopiriform nucleus (DEPN) and piriform (Pir) cortex was revealed using immunohistochemistry (**A**). The densities of c-fos-positive nuclei in saline- and PTZ (50 mg/kg)-treated mice were quantified (**B**,**C**). For saline group, *n* = 3 per genotype; for PTZ group, *n* = 5 per genotype. Results are mean ± SEM. Asterisks indicate significant differences (* *p* < 0.05; ** *p* < 0.01; *** *p* < 0.001). Bar = 200 μm.

**Figure 5 ijms-21-04921-f005:**
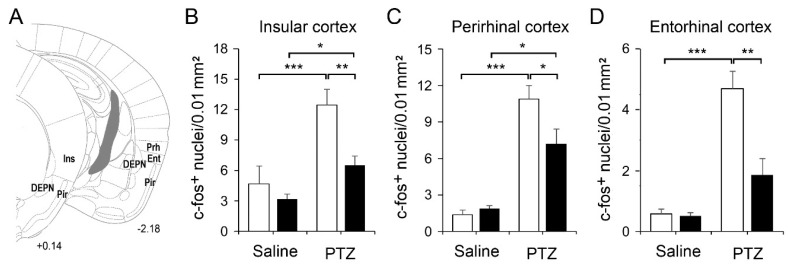
Schematic diagrams show the DEPN and nearby structures including the piriform (Pir) insular (Ins), perirhinal (Prh) and entorhinal (Ent) cortices (**A**). The densities of c-fos-positive nuclei were quantified (**B**–**D**). For saline group, *n* = 3 per genotype; for PTZ group, *n* = 5 per genotype. Results are mean ± SEM. Asterisks indicate significant differences (* *p* < 0.05; ** *p* < 0.01; *** *p* < 0.001).

**Figure 6 ijms-21-04921-f006:**
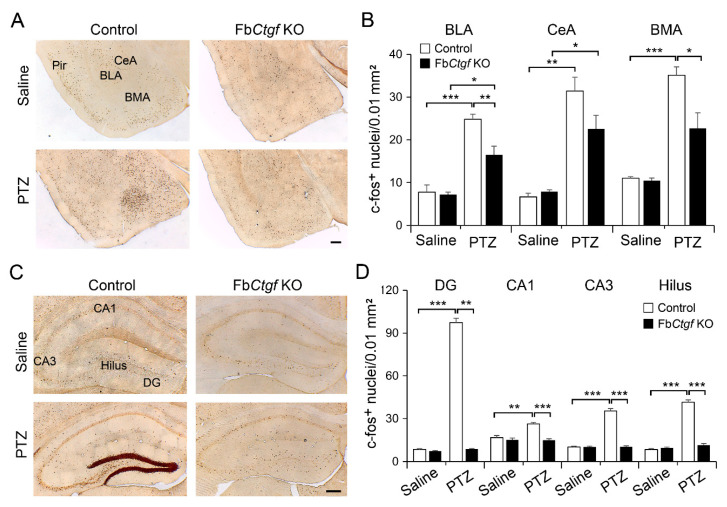
Expression of c-fos in the amygdala and hippocampus. c-fos was expressed in the basolateral amygdala (BLA), central amygdala (CeA) and basomedial amygdala (BMA) (**A**) and the densities of c-fos-positive nuclei were quantified (**B**). The expression of c-fos in the hippocampus including the DG, hilus, CA1 and CA3 regions (**C**) was quantified (**D**). For saline group, *n* = 3 per genotype; for PTZ group, *n* = 5 per genotype. Results are mean ± SEM. Asterisks indicate significant differences (* *p* < 0.05; ** *p* < 0.01; *** *p* < 0.001). Bar = 200 μm.

**Figure 7 ijms-21-04921-f007:**
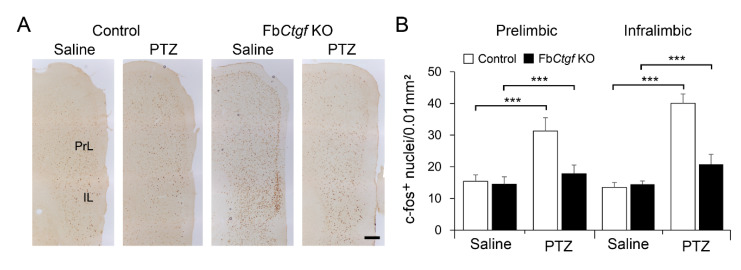
Expression of c-fos in the prelimbic and infralimbic cortices (**A**) and the densities were quantified (**B**). For saline group, *n* = 3 per genotype; for PTZ group, *n* = 5 per genotype. Results are mean ± SEM. Asterisks indicate significant differences (*** *p* < 0.001). Bar = 200 μm.

**Figure 8 ijms-21-04921-f008:**
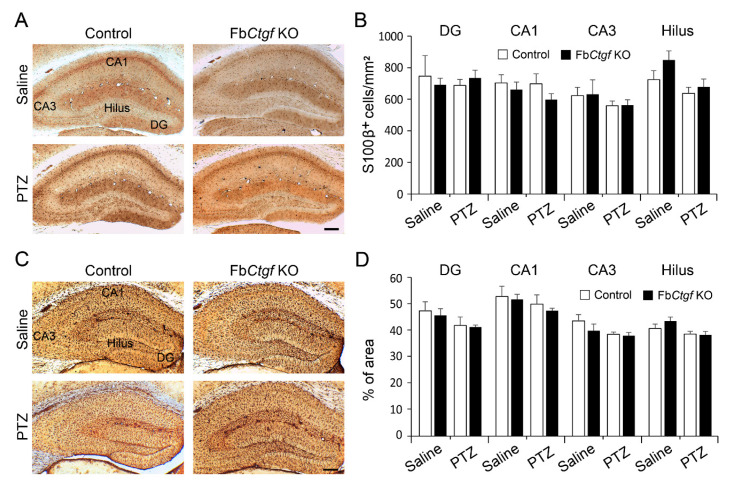
Status of astrocytes in the hippocampus. The density of S100β-positive astrocytes (**A**) and the area of glial fibrillary acidic protein (GFAP)-positive signals (**C**) in the hippocampus of saline- and PTZ (50 mg/kg)-treated mice were measured (**B**,**D**). For saline group, *n* = 3 per genotype; for PTZ group, *n* = 5 per genotype. Results are mean ± SEM. Bar = 200 μm.

**Figure 9 ijms-21-04921-f009:**
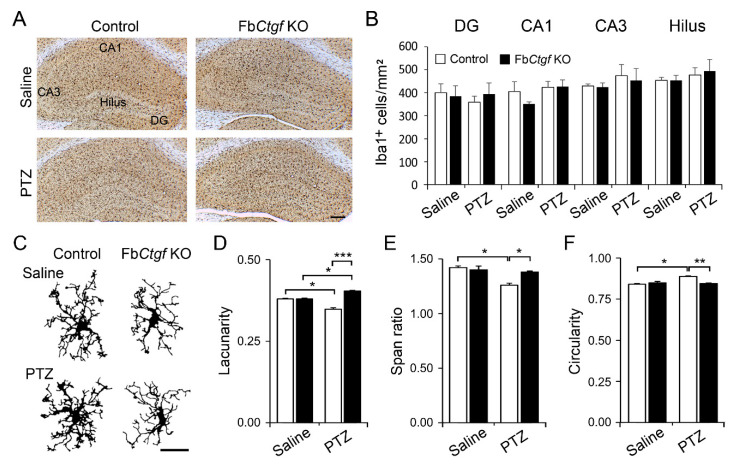
The densities of ionized calcium-binding adapter molecule 1 (Iba1)-positive microglia in the hippocampus (**A**) were measured (**B**). The morphology of Iba1-positive microglia in the DG was reconstructed (**C**) and the morphometric features were measured (**D**–**F**). For saline group, *n* = 20 cells from 3 control and 15 cells from 3 KO mice; for PTZ group, *n* = 17 cells from 5 control and 46 cells from 5 KO mice. Results are mean ± SEM. Asterisks indicate significant differences (* *p* < 0.05; ** *p* < 0.01; *** *p* < 0.001). Bar = 200 μm in (**A**) and 50 μm in (**C**).

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
