# Peer review of "Mice Lacking Connective Tissue Growth Factor in the Forebrain Exhibit Delayed Seizure Response, Reduced C-Fos Expression and Different Microglial Phenotype Following Acute PTZ Injection"

_ijms, 2020, doi:10.3390/ijms21144921_

Round 1
Reviewer 1 Report
Review of Siow et al. manuscript for International Journal of Molecular Sciences
Title: “Mice lacking connective tissue growth factor in the forebrain exhibit delayed seizure response, reduced c-fos expression and different microglial phenotype following acute PTZ injection”
Summary: In the article by Siow et al., the authors investigate a potential role of the endopiriform nucleus in seizure activity using a previously generated Ctgf conditional knockout mouse (crossed with Emx1-Cre to limit knock out to forebrain excitatory neurons – primarily cortex and claustrum/endopiriform). Using PTZ injection to induce seizures (a well-established seizure model), the authors first assess any behavioral phenotypic differences in seizure related activity between FbCtgf-KO and littermate controls. No significant differences were observed, but at the PTZ50 dose a delay in maximum seizure score was observed in the KO mice (needs to be statistically analyzed). The authors then analyze cFos expression of neuronal activity, astrocytic activity (S100 and GFAP), and activated microglia (Iba1) in multiple brain regions related to seizure activity and the endopiriform nucleus. The most startling finding was the complete lack of cFos expression in the dentate gyrus of the hippocampus in the KO mice after seizure compared to robust cFos expression in the controls. Additional regions belonging to traditional limbic system also showed diminished cFos, including regions of the amgydala, entorhinal and perirhinal cortex, and the endopiriform nucleus. Though no gross differences were observed in astrocytes or microglia in the dentate gyrus, there was a significant morphology difference in activated microglia in the DG of KO mice following seizure compared to control mice after seizure. The results of the study are well described and documented but require a few additional analyses before the manuscript is suitable for publication.
Specific Comments:
- Regarding Figure 1 showing expression of Ctgf in mice. It is not clear if this is in situ labeling (RNA) or immunohistochemistry (protein). Please label as appropriate and describe in the Results, Figure Legend and Methods.
- Additionally, regarding Figure 1, a lower magnification image of the endopiriform nucleus is important. Based on in situ expression from the Allen Brain Atlas, Ctgf is expressed in both the dorsal claustrum and the dorsal endopiriform nucleus (but NOT the ventral claustrum) as shown in the labeled images below (link to the specific data on Allen Brain Atlas: https://mouse.brain-map.org/experiment/show/1183). This is essential to illustrate. For reference, please see Smith et al, J Comp Neurol 2019 (https://pubmed.ncbi.nlm.nih.gov/30225888/). Also, can the authors comment on whether expression is in the dorsal, intermediate and ventral endopirifom or just the dorsal?? My analysis of the Allen Brain Atlas indicates that it is only in the dorsal endopiriform.
- For Figure 2, the authors need to run a 2-way repeated measures ANOVA on Panel E if they want to claim anything about the delay in seizure score (as they do in the abstract). Also, though nothing else was significant, reporting statistical values for the non-significant behavioral findings would be very useful. Especially under Panel B, where latency at PTZ 60 looks like it may have been close to significance. Also, it is not clear why PTZ 60 was not analyzed in Panel D.
- For Figure 3, the authors should show images of and do quantitative analysis of cFos expression in infralimbic cortex. Dorsal endopiriform is most strongly connected to infralimbic cortex and entorhinal cortex (analyzed in Figure 4, which had a significant effect in the KO mice). For reference on the connectivity of endopiriform nucleus, see Watson et al., J Comp Neurol, 2017 (https://www.ncbi.nlm.nih.gov/pmc/articles/PMC4980296/).
The authors may also consider analyzing Layer 6b, the other major locus of Ctgf expression as shown in Figure 1. How is cFos expression altered in this cortical layer compared to others??
- For Figure 4, the authors should show images of cFos data for the endopiriform nucleus (as done for other brain regions in Figures 3 and 5) as this nucleus is the focus of the entire paper.
- Figure 5: No comment here other than that panel B is absolutely amazing.
- Regarding morphometric data in Figure 8, it seems likely that microglial response is not a cause of the reduced neural activity, but an effect that neural activity in the DG was considerably lower. Consider amending the Discussion related to this point. This is also interesting that the KO mice were having a seizure independent of the dentate gyrus, indicating an entirely cortically driven seizure focus (as evident in the cortical cFos data from Figure 3).

Author Response
To reviewer 1: Many thanks to the reviewer, we have made proper changes in red in our revision according to your comments.
Comment 1: Regarding Figure 1 showing expression of Ctgf in mice. It is not clear if this is in situ labeling (RNA) or immunohistochemistry (protein). Please label as appropriate and describe in the Results, Figure Legend and Methods.
Response 1: Thanks for the reminder. We did Ctgf immunohistochemistry to reveal the expression pattern of Ctgf protein in the brain. We added this in the Results, Figure legends and Methods, accordingly.
Comment 2: Additionally, regarding Figure 1, a lower magnification image of the endopiriform nucleus is important. Based on in situ expression from the Allen Brain Atlas, Ctgf is expressed in both the dorsal claustrum and the dorsal endopiriform nucleus (but NOT the ventral claustrum) as shown in the labeled images below (link to the specific data on Allen Brain Atlas: https://mouse.brain-map.org/experiment/show/1183). This is essential to illustrate. For reference, please see Smith et al, J Comp Neurol 2019 (https://pubmed.ncbi.nlm. nih.gov/30225888/). Also, can the authors comment on whether expression is in the dorsal, intermediate and ventral endopirifom or just the dorsal?? My analysis of the Allen Brain Atlas indicates that it is only in the dorsal endopiriform.
Response 2: Thanks for the comment. We prepared lower magnification images of Ctgf-expressing brain areas including the layer VIb and dorsal endopiriform nucleus (DEPN) in our revised version. We agreed the reviewer’s comment that Ctgf is expressed in the dorsal endopiriform nucleus and not in the claustrum and ventral endopiriform nucleus. In our revision, we used DEPN to replace EPN.
Comment 3: For Figure 2, the authors need to run a 2-way repeated measures ANOVA on Panel E if they want to claim anything about the delay in seizure score (as they do in the abstract). Also, though nothing else was significant, reporting statistical values for the non-significant behavioral findings would be very useful. Especially under Panel B, where latency at PTZ 60 looks like it may have been close to significance. Also, it is not clear why PTZ 60 was not analyzed in Panel D.
Response 3: Thanks for the suggestion. We did two-way repeated measurements ANOVA on panel E. We found interactions between genotypes and time after PTZ treatment between 3-8 minutes (F(5,105) = 4.178; p = 0.008). For the latencies, we apology for the mistakes, in our original submission, we put the results of tonic-clonic (t.c.) in panel A, however it should be the results of generalized clonus (g.c.). Mistakes also occurred in panels B and C. We corrected all the mistakes in the Figure 2 and put p values in the text in our revision. We did not include PTZ 60 in panel D, due to the lethality of animals.
Comment 4: For Figure 3, the authors should show images of and do quantitative analysis of cFos expression in infralimbic cortex. Dorsal endopiriform is most strongly connected to infralimbic cortex and entorhinal cortex (analyzed in Figure 4, which had a significant effect in the KO mice). For reference on the connectivity of endopiriform nucleus, see Watson et al., J Comp Neurol, 2017 (https://www.ncbi.nlm.nih.gov/pmc/ articles/PMC4980296/). The authors may also consider analyzing Layer 6b, the other major locus of Ctgf expression as shown in Figure 1. How is cFos expression altered in this cortical layer compared to others??
Response 4: Thanks for the suggestion. We added the quantitative analysis of cFos expression in the prelimbic cortex and infralimbic cortex as a new figure in our revision. Similar to the pattern in the hippocampus, PTZ elevated the expression of c-fos in control, but not in FbCtgf KO mice. In the layer VIb, the expression of c-fos was elevated following PTZ treatment, but we did not observe layer-specific change.
Comment 5: For Figure 4, the authors should show images of cFos data for the endopiriform nucleus (as done for other brain regions in Figures 3 and 5) as this nucleus is the focus of the entire paper.
Response 5: Thanks for the suggestion. We put images of c-fos expression in the dorsal endopiriform nucleus and piriform cortex in our revision.
Comment 6: Regarding morphometric data in Figure 8, it seems likely that microglial response is not a cause of the reduced neural activity, but an effect that neural activity in the DG was considerably lower. Consider amending the Discussion related to this point. This is also interesting that the KO mice were having a seizure independent of the dentate gyrus, indicating an entirely cortically driven seizure focus (as evident in the cortical cFos data from Figure 3).
Response 6: Thanks for the comment. Yes, we think the PTZ-induced seizures in KO mice might be independent of c-fos expression in the hippocampus but relevant to the neuronal activity in the cortex.
Reviewer 2 Report
this is a well planned and executed study. I have only minor comments:
- I'm puzzled by the index "latency to death" since authors state that animals were sacrificed at 2h after PTZ injection;
- pannels showing immunostainings could be increased;
- Fig. 3 - it seems that staining intensity is layer-specific; neurons in which layer were counted?
- Fig. 5 - its difficult to see at this magnification, but I have an impression that PTZ induced in KO mice expression in a population of cells that is dispersed troughout tha CA1-CA3, DG and hilus; If so, what would be indentity of these cells
- page 10, line 257 - should be "respond"
Author Response
To review 2 Many thanks to the reviewer, we have made proper changes in blue in our revision according to your comments.
Comment 1: I'm puzzled by the index "latency to death" since authors state that animals were sacrificed at 2h after PTZ injection;
Response 1: We observed mice treated with PTZ for 30 minutes and measured the latency to death. We sacrificed the survived mice 2h after PTZ injection.
Comment 2: pannels showing immunostainings could be increased;
Response 2: Thanks for the suggestion. We added more immunostainings in our revision.
Comment 3: Fig. 3 - it seems that staining intensity is layer-specific; neurons in which layer were counted?
Response 3: Thanks for the comment. We measured the c-fos-positive nuclei in the upper cortical layers (layers II-III).
Comment 4: Fig. 5 - its difficult to see at this magnification, but I have an impression that PTZ induced in KO mice expression in a population of cells that is dispersed throughout the CA1-CA3, DG and hilus; If so, what would be the identity of these cells?
Response 4: Thanks for the comment. We did not identify the c-fos-expressed cells in the present study. We should include this experiment in our future work.
Comment 5: page 10, line 257 - should be "respond"
Response 5: Thanks, we corrected it in our revision.